



# Influence of uncertain identification of triggering rainfall on the assessment of landslide early warning thresholds

David J. Peres[1], Antonino Cancelliere[1], Roberto Greco[2], and Thom A. Bogaard[3]

[1]Department of Civil Engineering and Architecture, University of Catania, Catania, Italy
[2]Dipartimento di Ingegneria Civile Design Edilizia e Ambiente, Università degli Studi della Campania Luigi Vanvitelli, Aversa (CE), Italy
[3]Water Resources Section, Faculty of Civil Engineering and Geosciences, Delft University of Technology, Delft, the Netherlands

*Correspondence to*: D J. Peres (djperes@dica.unict.it)

**Abstract**

Uncertainty in rainfall datasets and landslide inventories is known to have negative impacts on the assessment of landslide-triggering thresholds. In this paper, we perform a quantitative analysis of the impacts that the uncertain knowledge of landslide initiation instants have on the assessment of landslide intensity-duration early warning thresholds. The analysis is based on an ideal synthetic database of rainfall and landslide data, generated by coupling a stochastic rainfall generator and a physically based hydrological and slope stability model. This dataset is then perturbed according to hypothetical "reporting scenarios", that allow to simulate possible errors in landslide triggering instants, as derived from historical archives. The impact of these errors is analysed by combining different criteria to single-out rainfall events from a continuous series and different temporal aggregations of rainfall (hourly and daily). The analysis shows that the impacts of the above uncertainty sources can be significant. Errors influence thresholds in a way that they are generally underestimated. Potentially, the amount of the underestimation can be enough to induce an excessive number of false positives, hence limiting possible landslide mitigation benefits. Moreover, the uncertain knowledge of triggering rainfall, limits the possibility to set up links between thresholds and physio-geographical factors.

## 1. Introduction

Thresholds estimating rainfall conditions correlated to landslide occurrence are useful for landslide early warning systems (Guzzetti et al., 2007; Highland and Bobrowsky, 2008; Sidle and Ochiai, 2013). Commonly, thresholds are derived by empirical approaches based on the direct statistical analysis of historical rainfall series and landslide inventories, from which a line roughly separating triggering from non-triggering conditions is drawn. Among the various thresholds types, precipitation intensity and duration power-law thresholds (hereafter referred to as ID thresholds), introduced by Caine (1980), have been derived for many regions of the Earth, and are still considered as a valid empirical model (Caracciolo et al., 2017; Gariano et al., 2015; Peruccacci et al., 2017; Vennari et al., 2014), though they have several limitations (Bogaard and Greco, 2017). Thresholds derived for different geographical areas vary significantly, and some attempts have been made to find a rationale underlying this variability, by linking threshold parameters to physio-geographical and climatic features (Guzzetti et al., 2007, 2008). Nevertheless, rainfall and landslide data quality issues, reported in almost all of the papers on threshold determination, are known to potentially hamper the assessment of this link. As reported in many studies, the triggering instants available from real landslide inventories are imprecise. For instance, Guzzetti et al. (2007, 2008) reported that in a global database of 2626 landslides, the vast majority (68.2 %) had no explicit information on the date or the time of occurrence of slope failure; for most of the remaining events only the date of failure was known, and more precise information was available just for 5.1% of landslides. These issues are confirmed with reference to an updated dataset of landslides occurred in Italy (Peruccacci et al.,



2017). In this case triggering instants were available with high precision (minute or hour) only for the 37.3% of the data, being the day or part of it available for the majority (27.6% and 35.1%, respectively).

Other data artefacts include: *i*) rainfall measurement delays; *ii*) different criteria to identify rainfall events; *iii*) lack of completeness of landslide catalogues; *iv*) imprecise location of landslides, or precipitation measurements available at a

significant distance apart from the location of failure. Though there is a general agreement that these factors affect the accuracy of rainfall thresholds, a quantification of the influence of these data quality issues on landslide triggering thresholds has been carried out in the literature only partially. In particular, to the authors knowledge, only the effect of rain gauge location and of the density of rainfall networks (point *iv*) has been analysed (Nikolopoulos et al 2014), showing that the use of rainfall measured at some distance from debris flow location can lead to an underestimation of the triggering thresholds.

Quantitative assessments of the influence of the sources of errors listed above are difficult to be based on observational datasets, since it cannot be ensured that these are immune of errors. In this paper we capitalize on the synthetic rainfall-landslide data set of a preceding study (Peres and Cancelliere, 2014), to quantify the effects of the imprecise identification of triggering rainfall on the assessment and performances of landslide triggering thresholds. The dataset is in principle "error free" in the

sense that the instants of landslide triggering are exactly known, as well as the triggering rainfall time history. We then fictitiously introduce errors in the triggering instants and in the rainfall series based on realistic scenarios of landslide data retrieval, and analyse the effect on thresholds. These scenarios are combined with different criteria for event rainfall identification, and different aggregations of rainfall data (hourly and daily), so the effects of these other two sources of uncertainty are analysed as well. The synthetic data used for our analyses are based on characteristic for hillslopes in the

landslide-prone region of Peloritani Mountains, in North-eastern Sicily, Southern Italy.

## 2 Dataset: generation of synthetic rainfall and landslide data

The dataset built in Peres and Cancelliere (2014) has been used here as reference. Here we provide a basic description of the methodology used for its development, which includes the following steps:

- *Synthetic generation of hourly rainfall time series:* A seasonal Neyman-Scott Rectangular Pulses (NSRP)
stochastic rainfall model (Cowpertwait et al., 1996; Rodriguez-Iturbe et al., 1987; Rodríguez-Iturbe et al., 1987) is used for the generation of 1000-years of hourly rainfall data. The model is calibrated on approximately 9 years of hourly observations from the Fiumedinisi rain gauge located in the area (Fig. 1).

- *Computation of hillslope pressure-head response*: A two-state hydrological model is used for the computation of pressure head. State 1 and 2 are activated separately during rainfall events and during no-rain intervals,
respectively. Rainfall events are defined as a section of the rainfall series preceded and followed by no rainfall for a minimum time interval of 24 hours. Within state 1 the TRIGRS-v2 model (Baum et al., 2010) is applied, which is based on the Richards' equation for mono-dimensional vertical infiltration with a Gardner negative exponential soil water characteristic curve. This is the least simplified form of the Richards' equation for which an analytical solution has been derived so far. A leakage flux at the soil-bedrock interface is considered, assuming the vertical
hydraulic conductivity of the bedrock strata $c_D = 0.1$ times the saturated conductivity $K_S$ of the pervious soil layer. Within state 2 a linear reservoir water table recession model is activated to simulate sub-horizontal drainage, and is used to compute water table height at the beginning of the next passage to state 1. A linear reservoir scheme computes a drainage flow that depends on the water table level, determining a negative-exponential decay of pressure head at the bottom of the regolith layer, with recession constant $\tau_M$.

- *Derivation of virtual landslide occurrence times*: An infinite slope model to compute factor of safety $F_S$ for slope stability is applied. For this schematization, failure surface coincides with the regolith-bedrock interface. The time





instants at which a downward crossing of $F_S = 1$ occurs are assumed to be the instants at which landslides are triggered.

The data set is generated considering soil hydraulic and geotechnical properties reported in Tab. 1 that can be considered representative of hillslopes in the Peloritani Mountains landslide-prone area (see Fig. 1). This area has been affected several times by catastrophic shallow landslide phenomena in the past; including the 1 October 2009 disaster, which has been analysed and described in several studies (Cama et al., 2017; Schilirò et al., 2015, 2016; Schilirò et al., 2015; Stancanelli et al., 2017). A morphological analysis of the catastrophic landslides occurred on 1 October 2009, has shown that a reasonable value of the recession constant for the specific case study area is $\tau_M = 2.7$ days (Peres and Cancelliere, 2014). Nevertheless, for the purposes of this study, we focus our analysis mainly on the hypothetical case of no pressure head memory ($\tau_M = 0$), so to isolate the source of impact of uncertainty in identification of triggering rainfall events. In other words, in the "ideal" simulations described above, the only uncertainty present is that of rainfall intra-event intensity variability, which is relatively small, so that a landslide-triggering threshold expressed in terms of rainfall duration and intensity performs almost perfectly (Peres and Cancelliere, 2014). For completeness, we however present a secondary analysis, in which $\tau_M = 2.7$ days. Table 2 shows some characteristics of the 1000-year long synthetic databases.

## 3 Methodology

### 3.1 Simulation of triggering rainfall identification uncertainty

As already mentioned, the available triggering instants from real landslide inventories are seldom precise. On the other hand, the instants at which landslides are triggered are known exactly (on hourly resolution) for the ideal synthetic series, illustrated in previous section. We then introduce errors in the triggering instants by hypothesizing the way such an information may be retrieved from newspapers, and similar resources (blogs and fire brigades), which in fact are the main primary sources available to build landslide historical inventories (e.g., Guzzetti and Tonelli, 2004). We suppose that only the date of the landslide is reported, and so is done with some delay. For a landslide to be reported on day $D$, it has to be observed within a time interval that goes from the night preceding that day to the end of its working hours (the "observer day"). Then the user of the landslide archive (the analyser), makes an interpretation of the available information, i.e. chooses an instant of the reported day of landslide occurrence to seek for the triggering rainfall.

Based on the above reasoning, we simulate the errors induced by the use of these sources by distinguishing an observation day $D'$, that ends at hour $T_O$ of day $D$, and an analyser time, $T_A$ (Fig. 2). The $i$-th landslide observed at $t_i$ within the observation day $D'$, i.e. hours [$T_O - 24$ h, $T_O$] of day $D$, is assumed by the analyser to be triggered $T_A$ hours after the start of day $D$ (civil day $D$ starts at 00:00). The observer day is made of the hours in which observers can report a landslide on day $D$. We assume that the observer day is given by hours going from 6 pm of day $D$–1 to 6 pm of day $D$ ($T_O = 18$ h). The analyser time is the instant of landslide triggering as considered by who analyses the data (the "analyser") to derive landslide-triggering thresholds, counted from the beginning of day $D$. This way to process the data introduces a sampling error and a shift between the actual instant at which the generic landslide $i$ is triggered $t_i$ and that assumed by who analyses the data, $t'_i$. Hence the error for the $i$-th landslide is given by:

$$e_i = t'_i - t_i \qquad (1)$$

A positive error can be in general considered as more probable than a negative, since landslides are typically reported after some time they have occurred (Guzzetti et al 1997, 1998; Peres and Cancelliere, 2013). This, however, does not exclude the possibility of a significant number of negative errors, because of temporal shifts in rainfall data, as will be discussed later.





The two parameters can be set to simulate different realistic scenarios. We perform our analysis based on four scenarios (which include the "ideal" one), hereafter referred to as landslide information "reporting scenarios" (RS), and illustrated in **Fig. 2**:

1.  *Ideal scenario* RS0 ($T_O = 0$, $T_A = 0$; $e_i = 0$ for all landslides). This is the ideal scenario (described in Sect. 2), without errors, that is considered as a reference for measuring errors in landslide triggering instants in the database simulated

by the three following scenarios.

2.  *Small delay reporting* RS1 ($T_O = 18$ h, $T_A = 24$ h; $0 \le e_i \le 30$ hours). A landslide occurring within the interval from night hours of $D - 1$ until the evening of day $D$ (i.e. within the observers' day $D'$) will be reported at day $D$. Here we suppose that the analyst attributes the landslide *at the end* day $D$ ($T_A = 24$ hours), i.e. search the triggering event backwards from that instant.

3.  *Large delay reporting* RS2 ($T_O = 18$ h, $T_A = 48$ h; $0 \le e_i \le 54$ hours). This scenario is similar to the previous, but here larger errors are hypothesized. We suppose that the landslide occurring during the observers' day $D'$ is reported on day $D + 1$, which is also erroneously assumed by the analyser as the day at which the landslide was triggered. He then attributes the landslide at the end day $D + 1$ ($T_A = 48$ hours). These timing errors may also be likely when landslides occur on weekends.

4.  *Anticipated reporting* RS3 ($T_O = 18$ h, $T_A = 0$ h; $- 18 \le e_i \le 6$ hours): This case is the same of RS1, but here analyst searches the triggering event backwards *from the beginning* of day $D$, i.e. at 00:00 (instead that at 24:00).

Within the context of sampling errors, another point is related to the way *rainfall* data is collected, specifically for daily data manually measured until some decades ago. A significant amount of papers derive landslide triggering thresholds using daily rainfall data (Berti et al., 2012; Leonarduzzi et al., 2017; Li et al., 2011; Terlien, 1998). In an ideal situation rainfall intensity

should be aggregated from 00:00 to 23:59, i.e. over a "civil day", as illustrated in Fig. 3. With reference to manual collection of rainfall data, this requires that raingauge should be read at midnight of each day, which is an uncomfortable hour. Manual collection of daily data is usually carried out at easier hours. For instance, in Italy, where the widest source of information are the Hydrological Bulletins (locally known as *Annali Idrologici*), the operator would measure the rainfall collected in the rainfall bucket every day at 9:00 am. Thus, daily rainfall in a given day is the amount of rainfall occurred in the 24 hours preceding 9

am of the same day. As illustrated in Fig. 3, in this case the reported daily rainfall amounts can be dramatically different than actual (see also Caracciolo et al., 2017).

Identification of triggering rainfall is uncertain also because of the different criteria that one can apply to isolate rainfall events from a continuous time series – Tab. 3 lists a range of criteria adopted in the literature. Here we analyse how the different criteria can impact the identification of triggering rainfall, both in the case that uncertainty in the triggering instants is present

(datasets RS1-RS3) or not (dataset RS0).

The procedure we adopt for isolating events is as follows (see Fig. 4). First, a minimum rainfall threshold $s_{min}$ is applied to all rainfall pulses at the fixed temporal aggregation. This means that from the original series a new one is obtained, where precipitation pulses less than $s_{min}$ are replaced by zeros. In the sketch these pulses are colored in light gray. Afterwards, rainfall events are singled-out when separated by zero-rain intervals longer than $u_{min}$. This parameter is the most important parameter

for definition of rainfall events. With the aim of quantifying how the impact of the errors implied by the different reporting scenarios changes with rainfall identification criteria, various pairs of $s_{min}$ and $u_{min}$ have been set (see Sect. 3.3). The described algorithm defines the rainfall event regardless it is associated or not to a landslide. For attributing a rainfall event to a landslide, the cases where the triggering instant is within a dry or wet period, should be analysed separately. In the first case, the landslide is associated to the whole closest event occurring before the landslide, in the other case it is to the part of rainfall event

occurring before the triggering instant.



Finally, triggering rainfall identification uncertainty is simulated by combining the reporting scenarios, different parameters of the rainfall event identification algorithm, and three rainfall aggregation schemes (hourly, daily correct and daily shifted). This results in twenty-eight combinations for each recession constant value $\tau_M$ (see Tab. 4).

**3.2 Threshold definition, calibration and testing performance**

Seventeen different landslide-triggering threshold types based on rainfall characteristics have been proposed in the literature in the period 1970-2006 (according to the list reported at rainfallthresholds.irpi.cnr.it, last date accessed 11 Sept. 2017). In spite of this variety, the most widely used is the rainfall intensity-duration (ID) threshold, as 96 out of 125 (about 77 %) are of this type, if one includes equivalent rainfall depth-duration (ED) thresholds. Therefore our analysis adopts this threshold type, which may be defined as follows:

$$I = \alpha\, D^{-\beta} \tag{2}$$

where $I$ [L/T] is the mean rainfall event intensity, $D$ [L] is the rainfall event duration (both defined according to scheme of **Fig. 4**). $\alpha, \beta > 0$ are respectively the intercept and slope parameters of the threshold. ED thresholds are equivalent to IDs, since rainfall intensity $I$ is the ratio between event rainfall $E$ (the total depth of a rainfall event) and its duration $D$; so an they can be

converted in the ID type by just subtracting 1 to the exponent of duration.

The procedures for the identification of best threshold parameters have historically increased their complexity through time. Early works have considered lower boundary curves of the triggering events traced with subjective criteria (Caine, 1980). Then more objective procedures have been then proposed, still based on the triggering events only, such as the so-called frequentist method (e.g., Brunetti et al., 2010). Finally, more advanced approaches are currently used, and are derived from

the analysis of both triggering and non-triggering events. These procedures are more transparent than methods based on triggering events only, as the uncertainty of the thresholds can be assessed though indices based on the confusion matrix, that is, in terms of the count of true positives (TP), true negatives (TN), false positives (FP) and false negatives (FN) (**Tab. 5**). More importantly, these methods are also more robust, since the presence of non-triggering data points makes the choice of the threshold less sensitive to possible errors in the attribution of triggering rainfall event duration and intensity. Here we use

these recent methods, implicitly assuming that the impact of the uncertainty under analysis is likely to be higher on thresholds derived from procedures based on triggering rainfall only.

Best-thresholds can be calibrated by maximizing their performances expressed in terms of suitable metrics. One widely used metric is the True Skill Statistics (Ciavolella et al., 2016; Peres and Cancelliere, 2014; Staley et al., 2013) proposed by Peirce (1884) :

$$\text{TSS} = \frac{\text{TP}}{\text{TP+FN}} - \frac{\text{FP}}{\text{TN+FP}} \tag{3}$$

An apparently alternative approach is given by Bayesian analysis (Berti et al., 2012). Indeed this approach can be interpreted as a special case of the ROC analysis, since Bayesian a-posteriori probability equals the ROC-based Precision (PRE):

$$P(L|R) = \frac{P(R|L)P(L)}{P(R)} = \frac{\frac{\text{TP}}{\text{TP+FN}}\frac{\text{TP+FN}}{N_T}}{\frac{\text{TP+FP}}{N_T}} = \frac{\text{TP}}{\text{TP+FP}} = \text{PRE} \tag{4}$$

where:

$P(L|R) = $ probability of landslide occurrence given rainfall exceeding the threshold (*a posteriori* probability),

$N_T = $ total number of rainfall events (triggering and non-triggering),

$P(R) = (\text{TP} + \text{FP})/N_T = $ probability of rainfall events exceeding the threshold,

$P(L) = (\text{TP} + \text{FN})/N_T = $ (*a priori*) probability of landslide occurrence,

$P(R|L) = \text{TP}/(\text{TP+FN}) = $ probability of rainfall event exceeding the threshold, given that a landslide has occurred (known as

the *likelihood*).





Different papers discuss advantages and disadvantages of various indices proposed in natural-hazard forecasting, as one single index is not sufficient to fully describe the confusion matrix (Frattini et al., 2010; Murphy, 1996; Stephenson, 2000). Nevertheless, the choice of a single index is essential to keep the calibration procedure simple, i.e. a single-objective

optimization problem. Hence, we here calibrate thresholds by maximizing the TSS.

Once thresholds for each RS scenario are derived, the TSS and the confusion matrix in general provide a measure of the uncertainty inherent the data, as assessable by who derives the threshold, and is not aware of the errors that could be present in the data. On the other hand, it is also of interest to test how a threshold derived from erroneous data may perform when, after its determination, it is applied to precise monitored data, and thus mostly free of the errors present in the threshold

calibration data set. To do so, the calibrated thresholds are applied to the ideal data set. The performances in this test are indicative of the impacts of errors when thresholds are actually used.

## 4 Impact of errors on threshold calibration

### 4.1 Hourly data

Results relative to the use of hourly data are shown in Fig. 5, for a given separation algorithm ($s_{min}$ = 0.2 mm, $u_{min}$ = 24 h). For the reference dataset RS0, there is a negligible overlapping between triggering and non-triggering events (Fig. 5a). In fact in this case the best ID threshold ($I = 101\,D^{-0.80}$) performs almost perfectly, with a TSS of 0.99 (for $u_{min}$ = 24 h). The presence of small delay reporting errors (RS1), has little impacts on the position of the triggering rainfall points (Fig. 5b). Two rainfall events are shifted to a duration of 1 hour, which contributes to slightly flatten the threshold (decrease of $|\beta|$ to 0.7). When high

delay sampling errors are present (RS2), the effects may not be negligible as in the previous case, as more erroneous rainfall events are present, now also for significant durations (up to 24 h in the plot, Fig. 5c). These errors are difficult to be identified by an analyser, and thus their impact on threshold determination can be significant, and lead to a lower slope and intercept, i.e. an underestimation of the threshold, which changes to $I=19D^{-0.50}$ (reference is $I = 101\,D^{-0.80}$). The impact of these errors may be more dramatic when thresholds are assessed making use of triggering rainfall events only, following "traditional", less

robust, approaches.

Negative errors, introduced by an anticipation of the real landslide instant (RS3), can have very high impacts, as can be seen from the relative plot in Fig. 5d, and the loss of the correct position of many of the triggering points. The best threshold corresponds to TSS = 0.49, which reflects the high degree of uncertainty implied by this kind of data errors.

### 4.2 Daily data

Shallow landslides can be triggered by rainfall events that are only some hours long (Bogaard and Greco, 2016; Highland and Bobrowsky, 2008; Sidle and Ochiai, 2013), and various studies have shown that the impact of intra-event rainfall intensity variability can have a significant effect on landslide triggering (D'Odorico et al., 2005; Peres and Cancelliere, 2014, 2016). Hence, apart from the errors in the dataset, it is of interest to see how the passage from hourly to daily data may affect threshold determination. This can be done by comparing thresholds determined from the hourly and daily datasets.

Figure 6 shows the results of calibration obtained with correctly-aggregated daily rainfall data and $s_{min}$ = 5 mm/day, $u_{min}$ = 1 day. As can be seen from the plots, the impact of delayed reporting of landslides (errors RS1 and RS2) is less significant than with hourly data. In fact, though α and β are lower than those determined from hourly data, the threshold determined from daily data passes more or less in the same zone for durations in their range of validity, D > 1 day. This is because the smaller slope β in the log-log plane compensates the smaller intercept α. The effect of anticipating landslide time

location (RS3) has also here high impacts on the thresholds, Fig. 6d.





Figure 7 plots the results relative to daily rainfall data affected by a delay in the aggregation interval, as present in Italian datasets, and related to use of non-automatic rain gauges. The impacts of this systematic rainfall error can be high (Fig. 7a, b, and d). There is, however, the possibility that the errors due to rainfall aggregation and reporting landslide time interval compensate for each other, as in the case of scenario RS2 (delayed reporting of landslides), Fig. 7c.

### 4.3 Possible effects of rainfall separation criteria and antecedent rainfall

Figure 8 shows the results obtained by setting the parameters of rainfall event separation algorithms, in the (a) hourly, (b) daily correct and (c) daily shifted data cases. From the TSS values shown in Fig. 8a it can be seen that the impact of RS1 and RS2 errors increases with decreasing minimum interarrival value $u_{min}$. In the case of RS3 error differences obtained with different $u_{min}$ are not relevant, since the performances are poor in general (TSS around 0.5). In the case of daily data (Figs. 8 b and c), the importance of different criteria for separating events (values of the minimum daily rainfall threshold $s_{min}$), are relatively lower than in the hourly data case. Though differences in the TSS are not significant, this may not be true for the thresholds parameters, which can vary significantly. In fact, a higher $s_{min}$ results in higher thresholds, because of the removal of days of with below a given rainfall amount.

The behaviour related to hourly data, may be due to the fact that, by choosing lower $u_{min}$, events get generally shorter and more numerous, and thus it is more likely that a landslide event is attributed to only part of the actual triggering event. In this case the effect of preceding rainfall events cannot be neglected in general. In other words, our analysis suggests that the choice of the $u_{min}$ is crucial, and must be based on the timescales of the hydrological processes governing landslide triggering, in terms of long and short term responses (Iverson, 2000). This means that the effect of different criteria for rainfall separation is somehow related to that of antecedent precipitation. The effects of antecedent precipitation is specifically taken into account performing Monte Carlo simulations with $\tau_M = 2.75$ days (results shown in Fig. 9). For this simulation, no matter what is the rainfall separation time interval, the initial water table height measured from the bottom of the soil column is in general greater than zero, becoming negligible after a dry interval of $3\tau_M = 3 \times 2.75 = 8.5$ days. As can be seen, the results are qualitatively similar to the no-memory case; the main difference is that lower TSS are obtained for the added uncertainty due to antecedent conditions, and the thresholds are lower, since less event rainfall is in average needed to trigger a landslide due to non-zero initial conditions.

### 5 Impact of errors on threshold use

Thresholds determined based on historical datasets are then meant to be used within early warning systems, when, consequently, more detailed meteorological and landslide monitoring is set up. This means that the thresholds determined with real datasets, affected by errors, are then applied to high quality datasets, less suffering of the limitations and errors present in datasets used for threshold determination, not initially conceived for that specific purpose. This might induce to modify the thresholds in view of the new data, but this process can take several years. Hence, with the aim of determining which would be the consequences of building an early warning system with thresholds derived from historical data with errors, Fig. 10 shows a visual comparison between the thresholds determined in the various numerical experiments and the ideal hourly dataset, for results related to the hourly (Fig. 10a) and daily data sets (Fig. 10b). For sake of clarity, it may be worthwhile to remember that the dataset of triggering and non-triggering points has been used in calibrating the thresholds only in the RS0 scenario (no errors), with hourly data, and $u_{min} = 24$ h, $s_{min} = 0.2$ mm (the related threshold is shown in Fig. 10a as a thick black line). Thus the other thresholds are tested against this ideal dataset, which is not the one used for their calibration.

The plots show that the presence of errors can induce a significant variability of thresholds which is totally unrelated to the different characteristics of a site (i.e. the geomorphological, hydraulic, geotechnical and land use characteristics). This allows to draw the hypothesis that a significant part of the variability of landslide triggering-thresholds reported in literature may be

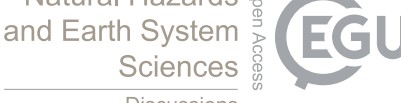



related to the sources of uncertainty here discussed. As a consequence, it is challenging to seek for links between the variability of physio-geographical characteristics and those of thresholds, as determined from different sites.

The presence of errors in the landslide dataset yields thresholds that are in general underestimated, i.e. lower than the correct ones. Many thresholds are significantly lower than the correct ones, and the number of false positives can be relatively high,

and not balanced by true positives. A good trade-off between correct and wrong predictions is essential for the success of an early warning system, since with an high number of false alarms the so-called cry-wolf effect may take place, inducing the populations not to take precautionary actions when warnings are issued (Barnes et al., 2007).

## 6 Conclusions

According to several studies, landslide inventories do not provide precise triggering instants information. In this paper, the

possible effects of this problem on the assessment of empirical thresholds for landslide initiation have been analysed and discussed, capitalizing on an ideal synthetic rainfall-landslide dataset generated by Monte Carlo simulation. To this aim, we have hypothesized reasonable scenarios of landslide information retrieval and interpretation, that can induce errors in the identification of instants of landslide occurrence. Moreover, we have analysed how the impact of reasonable scenarios may vary in dependence of rainfall aggregation (hourly or daily), and of rainfall event identification criteria.

The errors in the time instants can be, in an algebraic sense, positive or negative, according to whether the landslide is reported after its actual occurrence or before, respectively. Following literature, positive errors are more likely than negative, since it is typical that a landslide is reported some time after its actual occurrence. Our analyses have shown that if these errors are limited to less than one day, their impacts on the threshold may be relatively low; yet if the delay is higher, impacts can be significant. Negative errors, though less probable, can also exist, based on how an analyst interprets the information retrieved

from landslide historical archives. The impact of these errors can be dramatic, as the location of triggering-events in the $\log D$ – $\log I$ plane can be completely modified. Errors in landslide triggering instants lead to triggering events that are shorter than the actual ones, so that their effect is to induce an incorrect identification of triggering rainfall for short durations. For higher durations (>1 day), the location of triggering events seems to be more robust, except when negative errors are present. This behaviour induces a flattening of the PID thresholds (i.e. a lower slope $|\beta|$) and an underestimation of the position parameter

of the threshold (i.e. a lower intercept $\alpha$).

The impact of reporting errors can change significantly in dependence of the algorithm adopted for rainfall event identification. Specifically, a shorter "maximum dryness" interval for event separation induces an increase of the impacts of all kind of landslide time reporting errors.

When thresholds are determined from daily data, the data analyst has to be aware of possible shifts/delays in the rainfall

accumulation interval, that is if precipitation reported for a given day is the total amount occurred in a shifted period (e.g., within the 24 hours preceding 9 am of that day rather than before midnight). Such a kind of shift affects, for instance, the Italian Hydrological Annual Reports, the largest rainfall data collection in Italy. The impacts of these shifts are potentially dramatic.

Overall, the presence of reporting errors in landslide triggering instants brings to lower thresholds, making them less

suitable to set up of landslide early warning systems, as they can lead to a high number of false alarms, generating a misbelief by populations that are expected to benefit from their implementation. Similar effects have been found as a consequence of rainfall measurement uncertainty on thresholds (Nikolopoulos et al., 2014a). Just these two sources of errors – always present in observed datasets – are enough to generate an uncertainty in thresholds assessment that is of significant magnitude. These results bring to the conclusion that the uncertainty inherent the available data can jeopardize the possibility to find a physically

based rationale underlying the variability of empirical landslide-triggering thresholds across different sites. In other words, with the quality of current available data, attempts of relating thresholds to climate and other regional characteristics can be





very difficult. An improvement of landslide and rainfall monitoring – e.g. rainfall, soil moisture and landslide satellite data, as well as landslide data crowd-sourcing (Guzzetti et al., 2012; Strozzi et al., 2013; Wan et al., 2014) – may be a step forward for overcoming these problems. Once accurate rainfall-landslide data are available, standardized methodologies have to be implemented to derive the thresholds, in order to allow their comparisons and to link their variability to site-specific landslide
susceptibility factors.

**Aknowledgements** David J. Peres was supported by post-doctoral contract on "Studio dei processi idrologici relative a frane superficiali in un contesto di cambiamenti climatici" (Analysis of landslide hydrological processes in a changing climate), at University of Catania.

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





**List of figures**


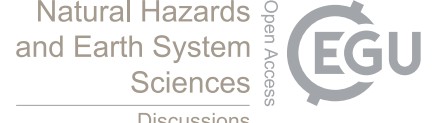



**List of tables**



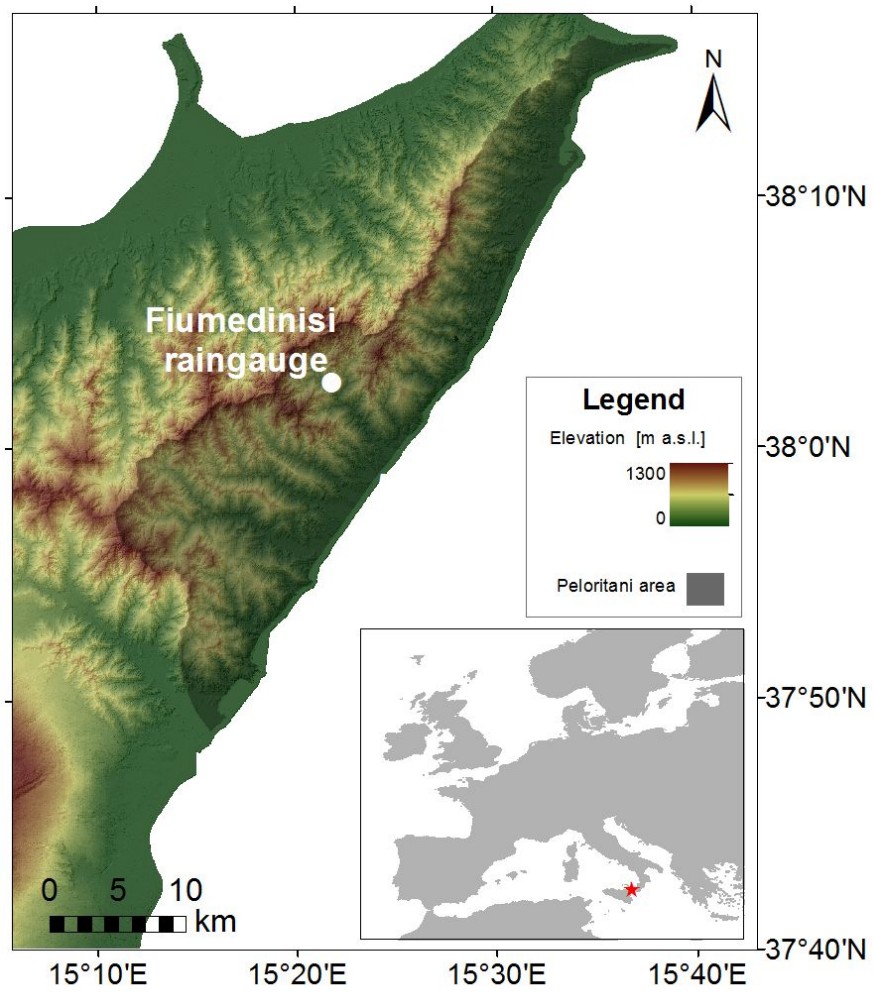

5    Figure 1: Location of the Peloritani Mountains area in Sicily, Italy, and of Fiumedinisi rain gauge.

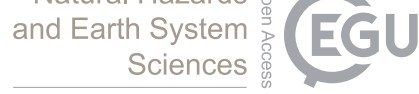



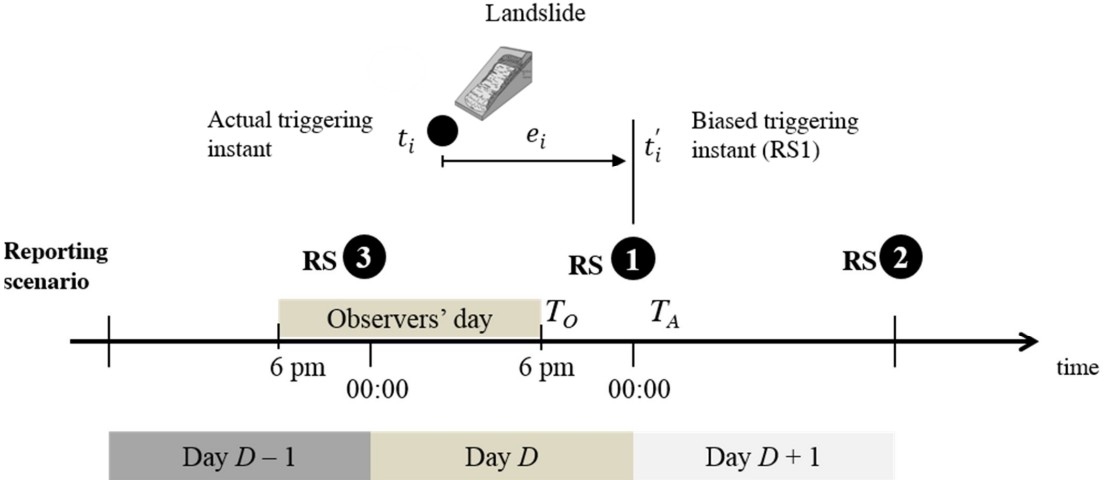

Figure 2: Sketch illustrating simulation of uncertainty in triggering instants likely present in landslide inventories built from newspapers or similar sources. The black numbered circles indicate one of the reporting scenarios (RS), each inducing random errors $e = t' - t$ in the landslide triggering instant. In particular, a landslide that occurs within the observers' day, is reported at day $D$ and attributed to the end of the same day (small delay reporting scenario, RS1) or to its beginning (anticipated reporting scenario, RS3). It can be reported also at day $D+1$ and thus attributed to the end of it (large delay reporting scenario RS2). These scenarios can be described in terms of two parameters: $T_O$ = the ending hour of observers' day, and $T_A$ = the triggering instant assumed by an analyser who interprets the newspaper-like information.

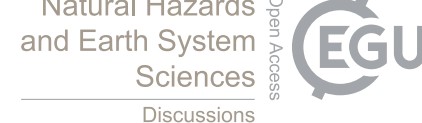



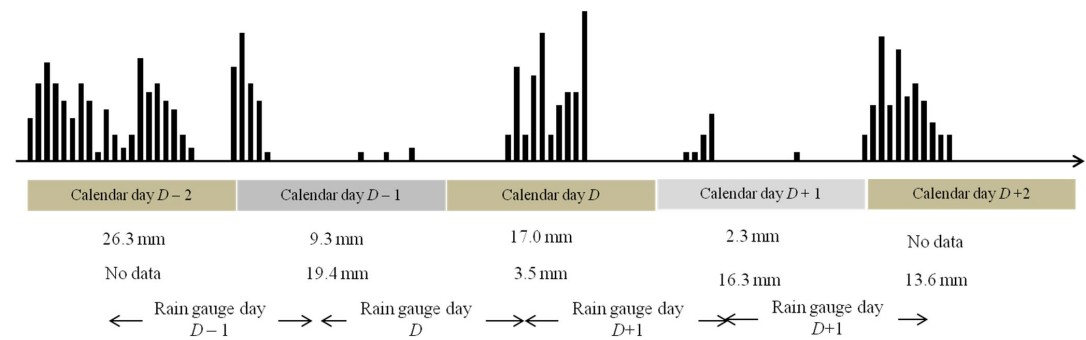

Figure 3: Aggregation of rainfall data from the hourly to the daily time scale: daily rainfall depths on the top row result from correct aggregation; those on the bottom row from shifted aggregation, as present in the Italian Hydrological Bulletins data.



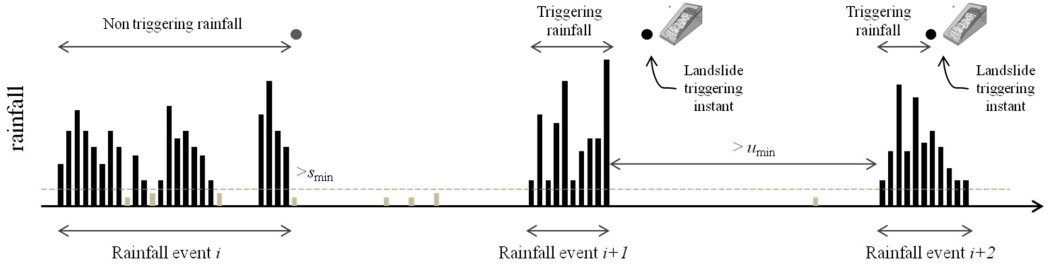

Figure 4: Sketch illustrating the algorithm for the identification of triggering and non-triggering rainfall events, and relative parameters $s_{min}$
5   and $u_{min}$. When a landslide is triggered in a dry period, it is attributed to the whole event preceding it; otherwise, only the part of the event preceding the landslide triggering instant is considered. For non-triggering rainfall (the first one in the sketch), duration and intensity are computed referring to the whole rainfall event.





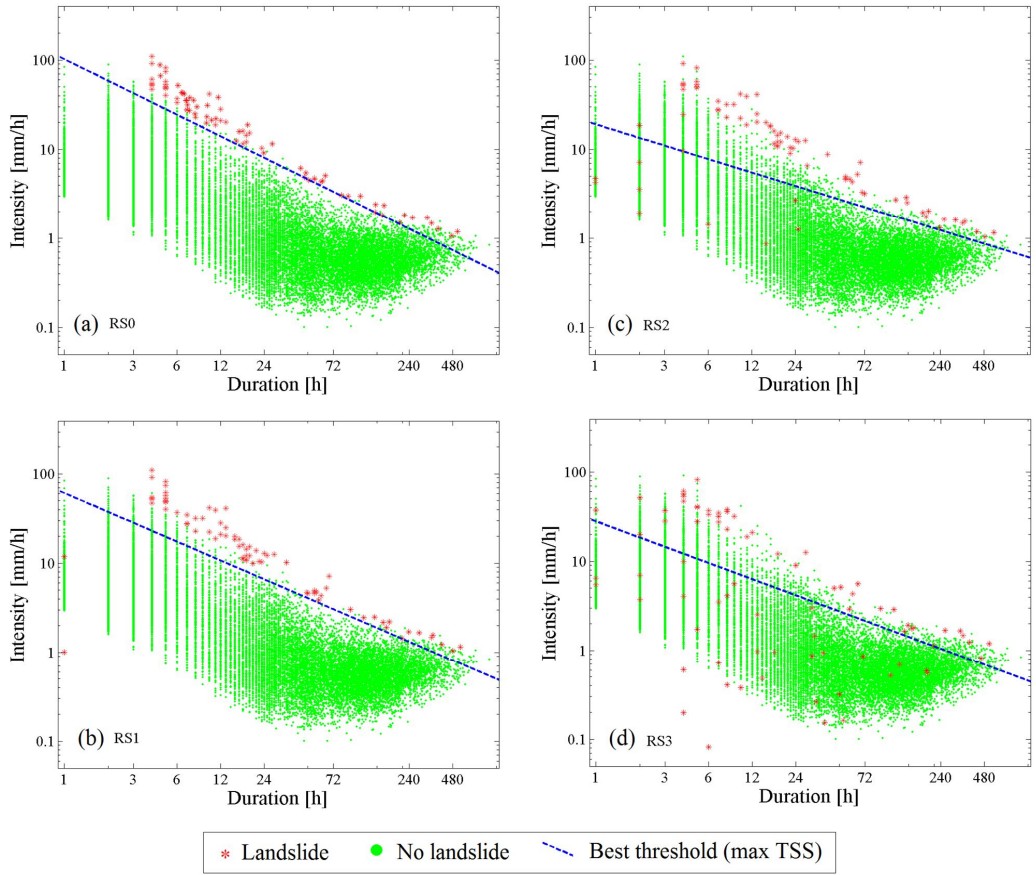

Figure 5 : Scatter plot, in the double-logarithmic rainfall duration-intensity plane, of triggering and non-triggering events for *hourly* data and
separation algorithm parameters $u_{min} = 24$ h, and $s_{min} = 0.2$ mm/hour. Thresholds correspond to the maximum performance in terms of True
Skill Statistic. The plots show outcomes relative to a) reference RS0, and b-d) various erroneous reporting scenarios (RS1, RS2, RS3).

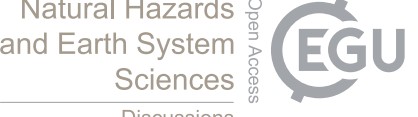



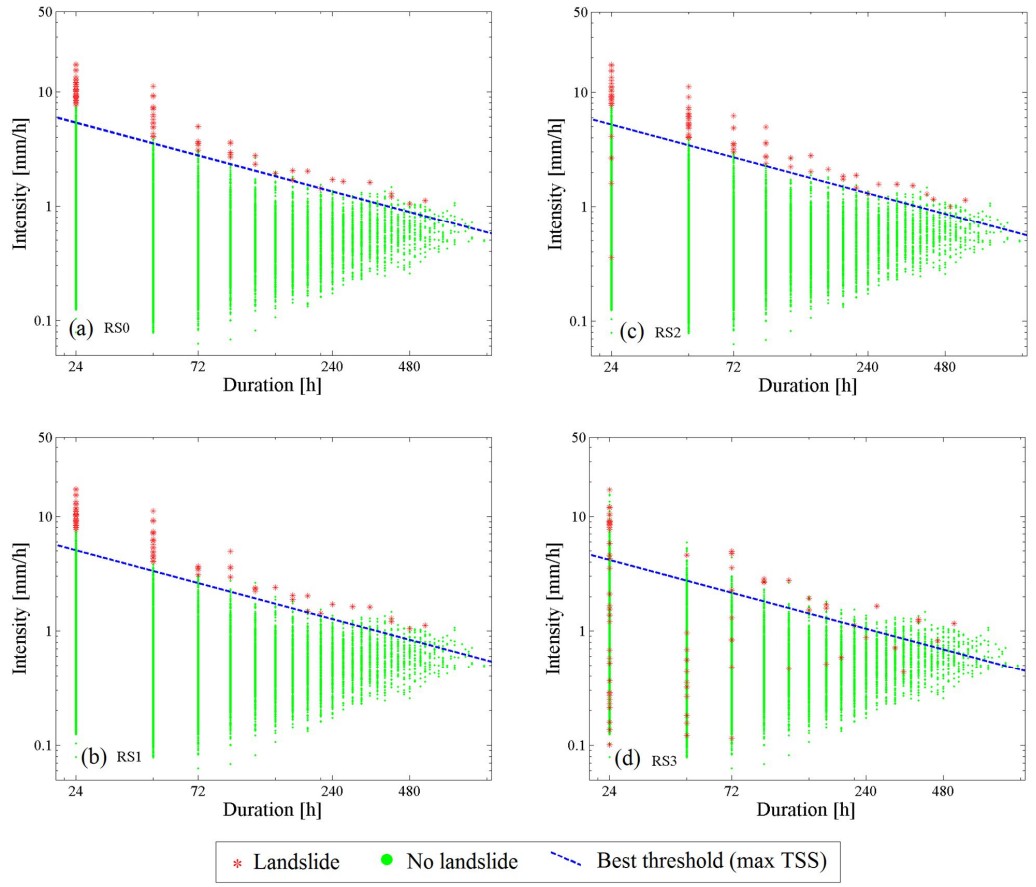

Figure 6: Scatter plot, in the double-logarithmic rainfall duration-intensity plane, of triggering and non-triggering events for *daily* data and
5   separation algorithm parameters $u_{min}$ = 1 day and $s_{min}$ = 0 mm. Thresholds correspond to the maximum performance in terms of True Skill
Statistic. The plots show outcomes relative to *a*) reference RS0, and *b-d*) various erroneous reporting scenarios (RS1, RS2, RS3).

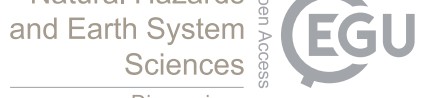



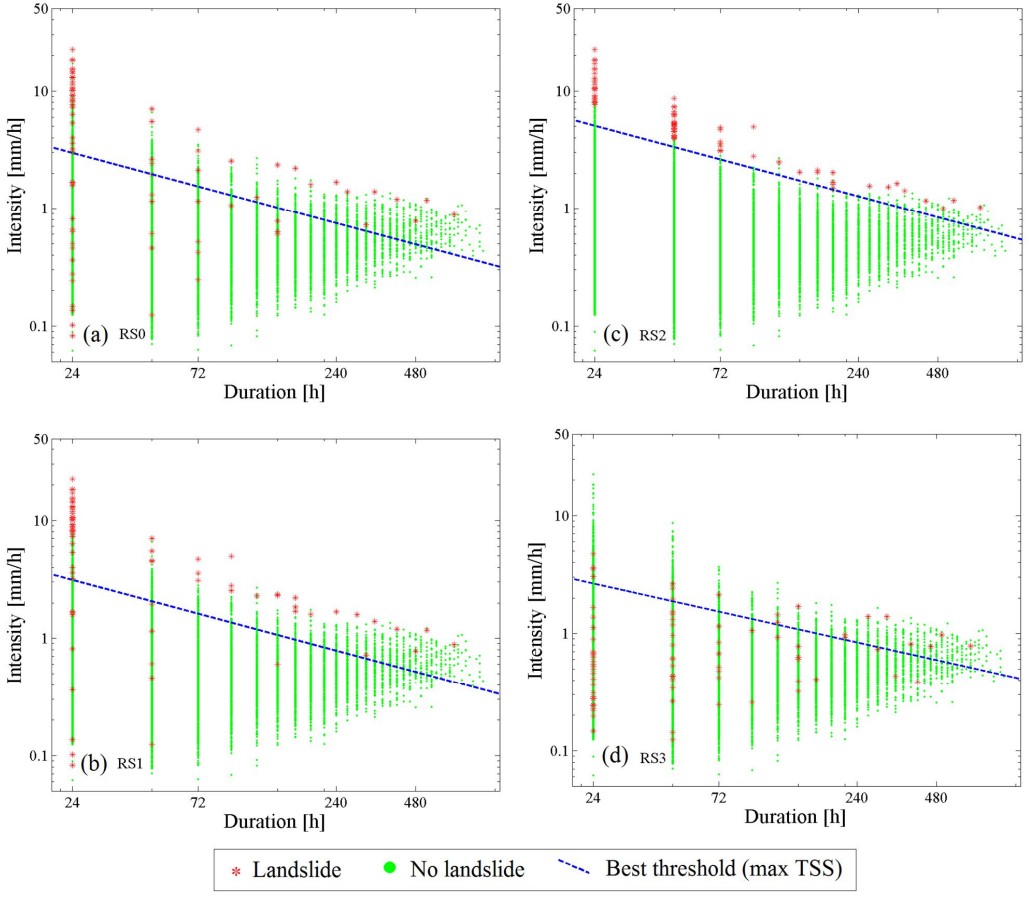

Figure 7: Scatter plot, in the double-logarithmic rainfall duration-intensity plane, of triggering and non-triggering events for *daily data with aggregation shift* as in the Italian rainfall databases. Separation algorithm parameters are: $u_{min}$ = 1 day and $s_{min}$ = 0 mm. Thresholds correspond to the maximum performance in terms of True Skill Statistic. The plots show outcomes relative to *a*) reference RS0, and *b-d*) various erroneous reporting scenarios (RS1, RS2, RS3).



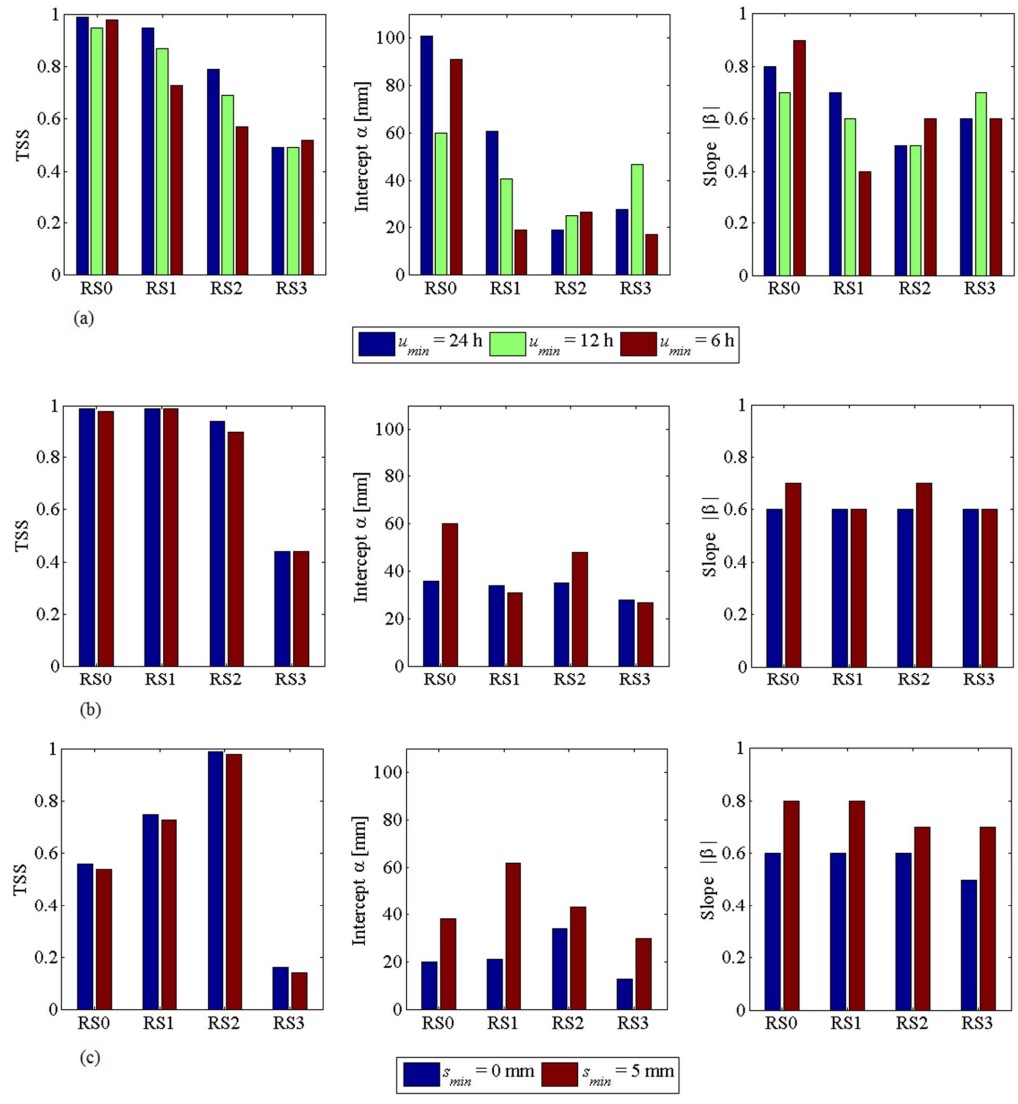

Figure 8: Threshold calibration results for all simulations. Plots show the value of the maximum TSS, and of best-threshold intercept $\alpha$ and
5  slope $\beta$ parameters, for different rainfall event identification algorithms and datasets: a) hourly resolution data, b) daily resolution and c)
daily resolution rainfall data with aggregation shift errors. Case of nulled effects of antecedent precipitation ($\tau_M = 0$).





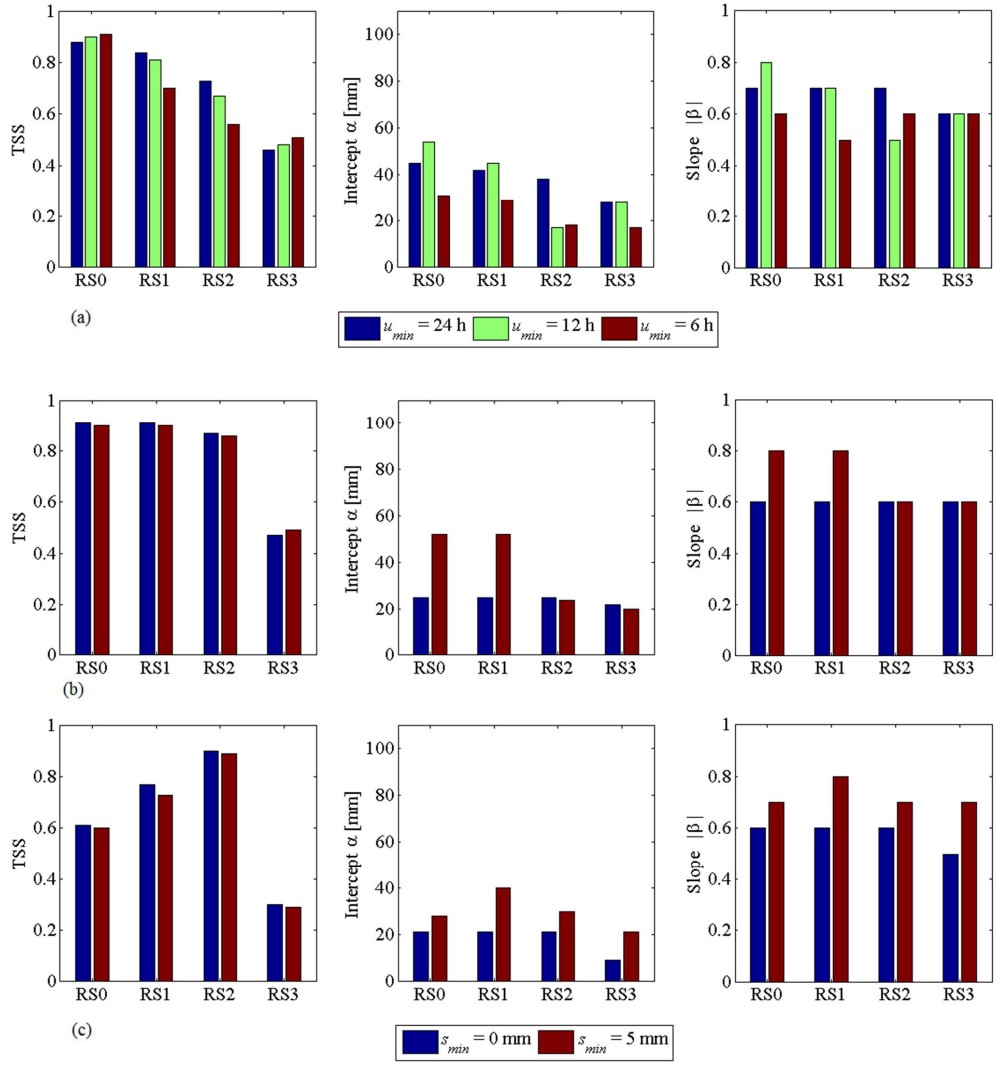

5  Figure 9: Same as Figure 8 but taking into account the presence of pressure head memory (recession constant $\tau_M$ = 2.7 days).

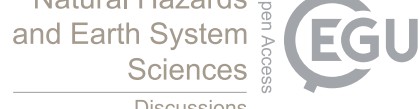



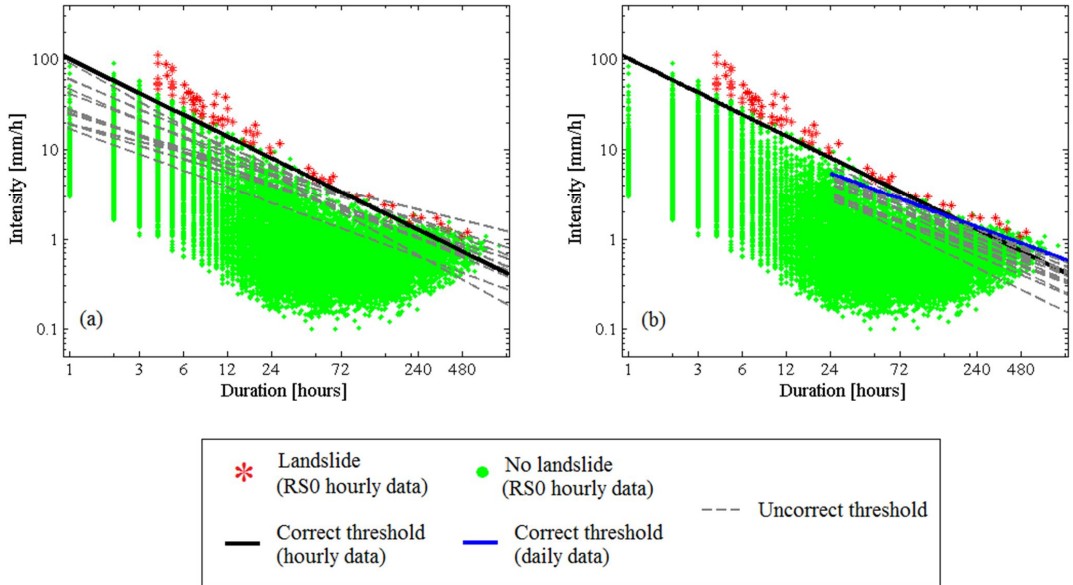

Figure 10: Comparison of thresholds, calibrated in the various scenarios and event identification parameters, with the correct hourly dataset. Thresholds determined with *a*) hourly and *b*) daily data (both correct and with aggregation shift), are distinguished. These plots are
5 representative of how thresholds calibrated with uncertain information of triggering rainfall data may perform in early warning systems that use high quality rainfall and landslide monitoring.



Table 1: Soil and morphological properties of a representative hillslope in the Peloritani Mountains Area, Sicily (after Peres and Cancelliere, 2014).

| Variable | Units | Value |
|---|---|---|
| Soil friction angle $\phi'$ | [°] | 37 |
| Soil cohesion $c'$ | [kPa] | 5.7 |
| Unit weight of soil $\gamma_s$ | [N/m$^3$] | 19000 |
| Saturated soil water content $\theta_s$ | [–] | 0.35 |
| Residual soil water content $\theta_r$ | [–] | 0.045 |
| Saturated soil hydraulic conductivity $K_s$ | [m/s] | 0.00002 |
| Saturated soil hydraulic diffusivity $D_0$ | [m$^2$/s] | 0.00005 |
| Gardner soil characteristic curve parameter $\alpha_0$ [*] | [1/m] | 3.5 |
| Soil depth $d_{LZ}$ | [m] | 2 |
| Terrain slope $\delta$ | [°] | 40 |
| Basal drainage leakage ratio $c_D$ [*] | [–] | 0.1 |

[*] See Baum and Godt, 2010 for details



Table 2: Some characteristics of the ideal Monte Carlo simulation dataset.

| Variable | Value |
| --- | --- |
| Number of simulated years | 1000 |
| Number of rainfall events | 19 826 |
| Number of landslide events for $\tau_M = 0$ | 81 |
| (return period) | 12.3 |
| Number of landslide events for $\tau_M = 2.7$ | 115 |
| (return period) | 8.7 |





Table 3: Some event identification algorithms found in the literature.

| Reference | Discretization | Algorithm parameters | |
|---|---|---|---|
| | | $s_{min}$ | $u_{min}$ |
| Pizziolo et al., 2008 | daily | 5 mm | 1 day |
| Berti et al., 2012 [*] | daily | 2 mm, *or* 1mm, *or* 2/3 mm | 1 day *or*, 2 days *or*, 3days |
| Rappelli, 2008 | hourly | 1 mm | 12 h |
| Melillo et al., 2015; Vessia et al., 2014 [**] | hourly | 0.2 mm | 3 h |
| | | | 6h |
| Saito et al., 2010 | hourly | 1 mm | 24 h |
| Segoni et al., 2014a, 2014b | hourly | 0 | $u_{min} = 10 \div 36$ h selected so that threshold performances were optimized |
| Brunetti et al., 2010; Peruccacci et al., 2017 | sub-hourly | 0 mm | 2 days (May-Sept) 4 days (Oct-Apr) |
| Peres and Cancelliere, 2014 | hourly | 0.2 mm | 24 h |
| Nikolopoulos et al., 2014b | hourly | 0.2 mm | 24 h |

(*) More precisely "the algorithm scans a rainfall time series and detect the rainfall events using a moving-window technique: a new event starts when the precipitation cumulated over $D_T$ days exceeds a certain threshold $E_T$, and ends when it goes below this threshold. For instance, if $D_T$ = 3 days and $E_T$ = 2 mm, the rainfall event starts when the cumulative rainfall exceeds 2 mm in 1, 2, or 3 days (that is if 2 mm are exceeded on the first day, the rainfall starts at day 1). Then, the rainfall event stops when it rains less than 2 mm in 3 days; the end of the event is defined as the last of the three days in which the rainfall is greater than zero". $D_T$ = 3 days and $E_T$ = 5 mm were chosen.

(**) The algorithm can be only approximately expressed in terms of $s_{min}$ and $u_{min}$. In particular, the algorithm additionally excludes "sub-events" having a total event rainfall below a seasonally variable threshold





Table 4: Set-up of the numerical experiments. Each set of algorithm parameters is considered for the four hypothesized landslide reporting-scenarios.

| Aggregation | Event identification algorithm parameters |
|---|---|
| Hourly | $u_{min}$ = 24 h, $s_{min}$ = 0.2 mm |
| | $u_{min}$ = 12 h, $s_{min}$ = 0.2 mm |
| | $u_{min}$ = 6 h, $s_{min}$ = 0.2 mm |
| Daily correct and daily shifted (Italian database) | $u_{min}$ = 1 day, $s_{min}$ = 0 mm/day |
| | $u_{min}$ = 1 day, $s_{min}$ = 5 mm/day |





Table 5: Confusion matrix for evaluation of landslide-triggering thresholds (assumed here to be of the ID type: $I = f(D)$).

|  |  | Actual | |
| --- | --- | --- | --- |
|  |  | Landslide (POS = TP + FN) | No landslide (NEG) (NEG = FP + TN) |
| Predicted | Landslide (POS'): $I \geq f(D)$ (POS' = TP + FP) | true positives, TP | false positives, FP |
|  | No landslide (NEG'): $I < f(D)$ (NEG' = FN + TN) | false negatives, FN | true negatives, TN |