# Peer review of "Influence of uncertain identification of triggering rainfall on the assessment of landslide early warning thresholds"

_Natural Hazards and Earth System Sciences, 2017_

## Referee Comment (RC1) · Anonymous Referee #1 · 10 Oct 2017

Review of the manuscript "Influence of uncertain identification of triggering rainfall on the assessment of landslide early warning thresholds" By David J. Peres, Antonino Cancelliere, Roberto Greco, and Thom A. Bogaard

General Comments The manuscript of Peres and co-authors entitled "Influence of uncertain identification of triggering rainfall on the assessment of landslide early warning thresholds" is an interesting well-structured and well-written manuscript that addresses a very important scientific question that is within the scope of NHESS. However, it needs some minor revisions prior to be published.

Specific Comments

1 - The exercise presented along the manuscript is based on synthetic data, which are easier to control and monitor. However, the exercise has the drawback of reporting a single ideal slope. So, there is also a matter of scale when we compare the obtained results with most rainfall thresholds reported in literature that were built to be applied and interpreted at the regional scale. May be this is not enough discussed along the manuscript.

2 - Within the simulation of uncertainty in triggering instant and the reporting of the landslide, authors establish the 'Observer's day' as lasting from the 6pm of Day D-1 to 6pm of the Day D. The explanation of this option is not clear. Although the reporting of a landslide in newspapers is usually delayed in relation to the actual triggering instant, the information about the timing of triggering may be quite precise namely in those cases where landslide generated severe human and/or economic damages. Apparently, this was not considered in the definition of the 'Observer's day'.

3 - Quite interesting, figures 6 a), 6 b) and 7 c) are very similar. Comparing figure 6a) and 6b) one can conclude that working at the daily scale the knowledge of exact timing of the landslide triggering is not essential, providing the reporting Day (D) is correct. In addition, when the daily rainfall depth is measured form 09:00 AM to 09:00AM it is clear that most of the rain that falls in the day D will be registered in the day D+1. Therefore, it is normal that threshold (c) corresponding to Scenario RS2 (Day D+1) in figure 7 is virtual similar to the Scenario RS1 (Day D) and RS0 (actual triggering instant) in figure 6. In the opinion of the reviewer, this topic should de discussed more in detail in the paper.

4 - Although this information is contained on Figures 8 and 9, the equations of thresholds could be provided in a summary table, allowing for a more easy comparison.

5 - When performing the exercise for the daily scale that is summarized in figure 6 and 7, a contradiction exists, between figures and text (page 6 line 35). on the assumed Smin. In figure caption it is referred Smin = 0 mm whereas in text is referred Smin = 5

mm.

6 - In figure 10 authors present the "correct thresholds". However, it is not given the information on the considered Umin and Smin parameters.

Technical corrections

In figure 2, the time scale should be respected. The position of 6pm in Day D and Day D-1 is not correctly scaled. Add the notation RS0 in figure 2.

Figure 3 The aggregation of data within figure 3 should be clearer. Rain gauge D+1 appear two times; why? The total amount of rain measured on calendar days and rain gauge days is not the same. Authors should acknowledge this difference and explain why.

Table 3 Some rainfall event identification instead of Some event identification.

Reference of the paper of Nikolopoulos et al needs to be corrected in reference list.

Page 2. Line 26 Rodriguez-Iturbe et al., 1987a, 1987b instead of Rodriguez-Iturbe et al., 1987; Rodríguez-Iturbe et al., 1987. Introduced a and b in the reference list.

Page 2, line 31 Baum and Godt, 2010, instead of Baum et al., 2010 ?

Page 3. Line 7 Schilirò et al., 2015a, 2015b, 2016; instead of Schilirò et al., 2015, 2016; Schilirò et al., 2015;

Page 3, line 38 Guzzetti et al 1997, 1998 are missing in reference list.

---

## Referee Comment (RC2) · Anonymous Referee #2 · 3 Nov 2017

GENERAL COMMENTS In this manuscript the authors investigate the effects of uncertain knowledge of the timing of landslide occurrence on the definition of intensity-duration rainfall thresholds. The study is based on synthetic rainfall data and virtual landslide events. Thresholds are defined using the True Skill Statistic as optimization criterion. The work is carried out for one ideal slope in the Peloritani Mountains in Sicily (IT). Overall the paper is well written, with a clear structure and objective. I believe it could benefit from some more elaborations on some of the aspects presented, mentioned here below. I recommend minor revisions before publication on the journal.

SPECIFIC COMMENTS

1 – On the line of what already mentioned by Anonymous Referee #1, the study is purely focused on one ideal slope and synthetic data. The authors could discuss how this might make the results transferable to a real situation, when regions are considered and heterogeneities come in to play. This with respect especially to the difference in the scale and the use of virtual landslides.

2 – The authors should report the total number of landslides as well as of non-triggering events considered. While this probably changes with the different parameters for the definition of the events, it would be useful to give an idea of the "robustness" of the results, that is whether the change of just few events among different scenarios would affect or not the threshold. Although the TSS considers both triggering and non-triggering events, the less the triggering events the more their relative importance on the definition of the threshold.

3 – The authors could elaborate more on how the threshold was defined, as the results are difficult to explain without this information. An example is the change going from the case shown in Figure 5a to 5b. The "two rainfall events shifted to a duration of 1 h" (line 18-19 page 6 in the text) cannot be responsible for the lowering of the threshold intercept or slope as they are not correctly captured by the threshold but are "missed". So either some other triggering events changed causing the decrease of the threshold or the threshold shouldn't have changed. All this is true unless the authors gave somehow weight also to the distance from the threshold. If being just below the threshold or well below the threshold makes a difference in the TSS, then yes those points could be responsible for the change and you should ignore this comment, but it would be helpful if the method would be explained.

4 – It seems that in general the points in the ID plane always move down (or left) in all the different scenarios. One would expect that sometime the landslides occur during intense rainfall storms and therefore including some extra hours actually could increase the intensity and duration.

5 – The authors could explain better how the different scenarios are then used and corresponding triggering events selected. In fact the scenarios are explained very well, but it is unclear how the events are then constructed. Is ei randomly selected for each virtual landslide within the range defined for each scenario? Are then the results shown only one possible realization? Or is the wrong timing always fixed to Ta (that is always midnight, either 0, 24 or 48)? In other words, is the triggering event always the one happening at midnight or the last one that happened just before then? That wouldn't be a very realistic case because one would either try to find out at least whether it was morning or afternoon, or choose the most intense event within the day (which would then result in an overestimation of the threshold, but probably would still better than taking midnight rain) or choose the typical timing of landslides. Also for an available database, not for all entries timing or at least part of the day would be unknown (for the example you report in line40page1 to line2page2, only 27.7% of the cases would fall in this case, of only day know)

6 – The case of the Italian rainfall dataset is presented in which precipitation for the day D is collected for the 24h preceding 9am of day D. Wouldn't one use this dataset by shifting it by one day? So that precipitation of day D is between 9am of day D and 9 am of day D+1? Surely there will still be some error as it still wouldn't match with the day definition, but this would probably be more meaningful.

---

## Author Response (AR1)

**Nat. Hazard Earth Syst. Sci. Discuss., https://doi.org/10.5194/nhess-2017-328**
Influence of uncertain identification of triggering rainfall on the assessment of landslide early warning thresholds, by David J. Peres, A. Cancelliere, R. Greco and T.A. Bogaard.

**Reply to Referee #1**

We thank the referee for reviewing our manuscript (MS). In the following we answer point by point to his constructive comments. Referee comments are in Times new roman black typesetting, our responses in Arial blue typesetting.

- *The authors*

**General Comments**

The manuscript of Peres and co-authors entitled "Influence of uncertain identification of triggering rainfall on the assessment of landslide early warning thresholds" is an interesting well-structured and well-written manuscript that addresses a very important scientific question that is within the scope of NHESS. However, it needs some minor revisions prior to be published.

Thanks again for the comments. Please see the following point by point replies.

**Specific Comments**

1 - The exercise presented along the manuscript is based on synthetic data, which are easier to control and monitor. However, the exercise has the drawback of reporting a single ideal slope. So, there is also a matter of scale when we compare the obtained results with most rainfall thresholds reported in literature that were built to be applied and interpreted at the regional scale. May be this is not enough discussed along the manuscript.

As correctly stated by the reviewer, to refer to synthetic data allows to isolate factors of uncertainty to test their influence on a issue of interest – on ID thresholds in the case of our manuscript.
It is certainly true that mostly thresholds are determined by analyzing rainfall-landslide data from multiple locations within a region. This means that the properties of unstable slopes change from landslide to landslide. Clearly this heterogeneity impacts on the performances of regional thresholds. This is a problem of empirical thresholds, and an additional source of uncertainty. To analyze this source of uncertainty in combination with that related to uncertain knowledge of triggering rainfall events, is out of the scope of our MS, and may be the scope of further research. A comment on this will be added to the text. An outlook in the conclusions mentioning this issue will be added as well.

2 - Within the simulation of uncertainty in triggering instant and the reporting of the landslide, authors establish the 'Observer's day' as lasting from the 6pm of Day D-1 to 6pm of the Day D. The explanation of this option is not clear. Although the reporting of a landslide in newspapers is usually delayed in relation to the actual triggering instant, the information about the timing of triggering may be quite precise namely in those cases where landslide generated severe human and/or economic damages. Apparently, this was not considered in the definition of the 'Observer's day'.

The 'Observer's day' is assumed as lasting from the 6pm of Day $D$-1 to 6pm of the Day $D$. This is justified by normal working hours at day $D$, plus the fact that what happens before in the night is reported in newspapers (and similar sources) from the next morning. The choice of 6pm rather than

another hour is quite arbitrary, but a different choice would not affect significantly our results. A small discussion on this will be added to the revised MS.

We agree with the referee that in real datasets there may be a portion of triggering instants known precisely. We preferred to do not consider "mixed scenarios" where small and big errors coexist in certain proportions. It may not be difficult to add those scenarios, but we believe that this would not add substantial changes to the conclusions of the manuscript, or even result in less clear findings. Mixed scenarios would produce impacts that are intermediate between two/three of the considered RS, depending on their percentage.

On this point we also refer to the reply to comment 5 of referee #2.

3 - Quite interesting, figures 6 a), 6 b) and 7 c) are very similar. Comparing figure 6a) and 6b) one can conclude that working at the daily scale the knowledge of exact timing of the landslide triggering is not essential, providing the reporting Day (D) is correct. In addition, when the daily rainfall depth is measured form 09:00 AM to 09:00AM it is clear that most of the rain that falls in the day D will be registered in the day D+1. Therefore, it is normal that threshold (c) corresponding to Scenario RS2 (Day D+1) in figure 7 is virtual similar to the Scenario RS1 (Day D) and RS0 (actual triggering instant) in figure 6. In the opinion of the reviewer, this topic should de discussed more in detail in the paper.

We agree with the reviewer about the comparison of Fig. 6a with Fig. 6b. Stronger comments will be added to the MS following the suggestion of the referee, though the point of the reviewer is already stated in the MS at two points: P6 L17-18; P8L17-18.
Relatively to comparison between Fig. 7c and Fig. 6b, we agree with the reviewer that there is a compensation of errors in this case, as already commented in the MS P7 L3-4. In the conclusions there it is also mentioned that this implicates that the analyzer should check if the original data are affected by this systematic error, and eventually compensate for it (P8 L29-30): "the data analyst has to be aware of possible shifts/delays in the rainfall accumulation interval"
However a more explicit suggestion for the "analyzer" to check and correct for this error will be added.

4 - Although this information is contained on Figures 8 and 9, the equations of thresholds could be provided in a summary table, allowing for a more easy comparison.

In the revised MS, Figures 8 and 9 will be replaced by Tables with the same information.

5 - When performing the exercise for the daily scale that is summarized in figure 6 and 7, a contradiction exists, between figures and text (page 6 line 35) on the assumed Smin. In figure caption it is referred Smin = 0 mm whereas in text is referred Smin = 5 mm.

The actual adopted value is $S_{min}$ = 0 mm. This will be corrected in the revised MS

6 - In figure 10 authors present the "correct thresholds". However, it is not given the information on the considered Umin and Smin parameters.

The actual adopted value is $S_{min}$ = 0.2 mm and $U_{min}$ = 24 h for the correct thresholds determined from hourly data, and $S_{min}$ = 0 mm and $U_{min}$ = 1 day for those determined from daily data. This will be specified in the caption of the figure.

**Technical corrections**

In figure 2, the time scale should be respected. The position of 6pm in Day D and Day D-1 is not correctly scaled. Add the notation RS0 in figure 2.

This will be fixed for the revised MS

Figure 3 The aggregation of data within figure 3 should be clearer. Rain gauge D+1appear two times; why? The total amount of rain measured on calendar days and raingauge days is not the same. Authors should acknowledge this difference and explain why.

This will be fixed. Improved figure and a more detailed caption will appear in the revised MS

Table 3 Some rainfall event identification instead of Some event identification.

This will be fixed for the revised MS

Reference of the paper of Nikolopoulos et al needs to be corrected in reference list.

This will be fixed for the revised MS

Page 2. Line 26 Rodriguez-Iturbe et al., 1987a, 1987b instead of Rodriguez-Iturbe et al., 1987; Rodríguez-Iturbe et al., 1987. Introduced a and b in the reference list.

This will be fixed for the revised MS

Page 2, line 31 Baum and Godt, 2010, instead of Baum et al., 2010 ?

This will be fixed for the revised MS. Correct is Baum et al., 2010. Mistake is in reference (third author missing)

Page 3. Line 7 Schilirò et al., 2015a, 2015b, 2016; instead of Schilirò et al., 2015,2016; Schilirò et al., 2015;

This will be fixed for the revised MS

Page 3, line 38 Guzzetti et al 1997, 1998 are missing in reference list.

This will be fixed for the revised MS

**Nat. Hazard Earth Syst. Sci. Discuss., https://doi.org/10.5194/nhess-2017-328**
Influence of uncertain identification of triggering rainfall on the assessment of landslide early warning thresholds, by David J. Peres, A. Cancelliere, R. Greco and T.A. Bogaard.

**Reply to Referee #2**

We thank the referee for reviewing our manuscript (MS). In the following we answer point by point to his constructive comments. Referee comments are in Times new roman (black) typesetting, our responses in Arial (blue) typesetting.

- *The authors*

**GENERAL COMMENTS**

In this manuscript the authors investigate the effects of uncertain knowledge of the timing of landslide occurrence on the definition of intensity duration rainfall thresholds. The study is based on synthetic rainfall data and virtual landslide events. Thresholds are defined using the True Skill Statistic as optimization criterion. The work is carried out for one ideal slope in the Peloritani Mountains in Sicily (IT). Overall the paper is well written, with a clear structure and objective. I believe it could benefit from some more elaborations on some of the aspects presented, mentioned here below. I recommend minor revisions before publication on the journal.

Thanks again to the referee for his comments, to which we reply in the "Specific Comments" section.

**SPECIFIC COMMENTS**

1 – On the line of what already mentioned by Anonymous Referee #1, the study is purely focused on one ideal slope and synthetic data. The authors could discuss how this might make the results transferable to a real situation, when regions are considered and heterogeneities come in to play. This with respect especially to the difference in the scale and the use of virtual landslides.

As we stated in the reply to referee #1, the use of synthetic data allows to isolate and test the effect of landslide triggering thresholds of single and controlled factors of uncertainty. When regions are considered, heterogeneities come in to play, which means additional sources of uncertainty in landslide threshold determination, which would make less clear the effects on the threshold of the source of uncertainty considered here. It is out of the scope of our MS to combine these two different sources of uncertainty. This will be more clearly stated in the revised paper, and discussed briefly.

2 – The authors should report the total number of landslides as well as of non-triggering events considered. While this probably changes with the different parameters for the definition of the events, it would be useful to give an idea of the "robustness" of the results, that is whether the change of just few events among different scenarios would affect or not the threshold. Although the TSS considers both triggering and non-triggering events, the less the triggering events the more their relative importance on the definition of the threshold.

Perhaps the information required by the referee is already shown in Table 2 of the MS: the number of landslides is 81 (115) and the number of non-triggering *rainfall* events is 19826 – 81 = 19745 (19711) for $\tau_M=0$ ($\tau_M$ = 2.7 days). These numbers do not change when different scenarios and different parameters for the definition of the rainfall events ($U_{min}$ and $S_{min}$) are applied. Hence the

effect on the TSS mentioned by the referee in not present, and does not affect the comparison of scenarios in terms of threshold determination and relative performances.

3 – The authors could elaborate more on how the threshold was defined, as the results are difficult to explain without this information. An example is the change going from the case shown in Figure 5a to 5b. The "two rainfall events shifted to a duration of 1 h" (line 18-19 page 6 in the text) cannot be responsible for the lowering of the threshold intercept or slope as they are not correctly captured by the threshold but are "missed". So either some other triggering events changed causing the decrease of the threshold or the threshold shouldn't have changed. All this is true unless the authors gave somehow weight also to the distance from the threshold. If being just below the threshold or well below the threshold makes a difference in the TSS, then yes those points could be responsible for the change and you should ignore this comment, but it would be helpful if the method would be explained.

We thank the referee for his suggestion to include more details on threshold determination. These will be added to the MS to better clarify how the TSS determines threshold position. However, in contrast to the referee's reasoning, Figure 5a and 5b differ for more than just the "two rainfall events shifted to a duration of 1 h" (line 18-19 page 6 in the text): the rainfall intensity and duration of generally *all* triggering events changes. Though these changes are relatively small, they still affect the position of the TSS-optimized thresholds. In other words, it is true that the TSS does not "weight the distance from the threshold", and so it is also true that only two points cannot be responsible for a significant change in threshold parameters and performances. It is rather the fact that *all the* triggering points in general change, though slightly. The figure below (Fig. R1) compares duration, depth and intensity of triggering events relative to the data in Fig. 5a ("no errors", RS0 hourly) and Fig. 5b ("with errors", RS1).

These details will be clarified in the revised MS (possibly with the addition of Fig R1).

[Figure]

Fig R1- Comparison of triggering event characteristics for scenarios RS0 and RS1 in the case of hourly data and $S_{min}$ = 0 and $U_{min}$ = 24h (cf. Fig 5a and 5b of the MS)

4 – It seems that in general the points in the ID plane always move down (or left) in all the different scenarios. One would expect that sometime the landslides occur during intense rainfall storms and therefore including some extra hours actually could increase the intensity and duration.

We thank the referee for this comment, which will help to clarify some aspects of the obtained results. In fact, while, as a consequence of errors in the triggering instants, the rainfall event duration $T$ may increase and the total rainfall depth $H$ too, their ratio (rainfall intensity $I$) seldom increases. This is well known from rainfall extreme event analysis – the so-called intensity-duration-frequency (IDF) curves have always negative slope (see, for instance Bogaard and Greco, 2017): this is related to the fact that the higher the duration, the lower the mean rainfall intensity tends to be. Again, Fig R1 can be looked at as a confirmation of this behavior. Moreover, the few events that may have an

increased $T$ and $I = H/T$, have a lower influence on threshold determination than the majority, which present decreased duration and intensity. This is not only because the events with increasing intensity are few, but also because the optimal threshold position is more sensitive to changes in the lower part of the cloud of triggering points (related to lower intensities), which partly mix up with the upper part of the non-triggering cloud. On the other side, the triggering points with increased intensity are usually not originally mixed up with the non-triggering cloud, and thus their change seldom determines a variation of maximum TSS.

These aspects will be shortly detailed in the revised manuscript.

*Refs.*

*Bogaard, T., Greco, R., 2017. Invited perspectives. A hydrological look to precipitation intensity duration thresholds for landslide initiation: proposing hydro-meteorological thresholds. Nat. Hazards Earth Syst. Sci. Discuss. 1–17. https://doi.org/10.5194/nhess-2017-241*

5 – The authors could explain better how the different scenarios are then used and corresponding triggering events selected. In fact, the scenarios are explained very well, but it is unclear how the events are then constructed. Is $e_i$ randomly selected for each virtual landslide within the range defined for each scenario? Are then the results shown only one possible realization? Or is the wrong timing always fixed to Ta (that is always midnight, either 0, 24 or 48)? In other words, is the triggering event always the one happening at midnight or the last one that happened just before then? That wouldn't be a very realistic case because one would either try to find out at least whether it was morning or afternoon, or choose the most intense event within the day (which would then result in an overestimation of the threshold, but probably would still better than taking midnight rain) or choose the typical timing of landslides. Also for an available database, not for all entries timing or at least part of the day would be unknown (for the example you report in line40 page1 to line2 page2, only 27.7% of the cases would fall in this case, of only day know)

The following may serve as clarification in respect to the above referee comments.

Within the RS1-RS3 scenarios, we assume that the analyzer attributes the landslide to a day. The most conservative option is to do so by searching the rainfall event backwards from the end of the day (24h in RS1 or 48h in RS2), the least conservative is to do it from the beginning (0h in RS3). With our scenarios we consider a range of possibilities respect to which real scenarios (datasets) may represent intermediate cases. Our objective is not to analyze the complex subjective process that the analyzer may adopt in searching for triggering rainfall. Indeed, subjective criteria have been criticized by several researchers (e.g. Berti et al, 2013; Vessia et al, 2014; Melillo et al., 2015 – papers already in MS references) in favor of automatic procedures, which are more objective and thus more scientifically sound. Interestingly, in the paper by Berti et al. (2013), an automatic algorithm is calibrated based on decisions taken by a group of "expert analyzers". Thus automatic procedures can proxy "expert analyzer" behavior, with the added advantage of reproducibility.

In order to clarify the origin of errors $e_i$, perhaps it is useful to more explicitly specify the difference between the real triggering date $t_i$ and the one at which the analyzer considers the landslide triggered $t_i'$ (that generally differs from $t_i$, because of the limited information available). It is the latter that is discretized at midnights; the former is determined by rainfall time history and thus is random. Thus errors $e_i = t_i' - t_i$ are implicitly random. The ranges indicated within brackets are the maximum and minimum values of the errors in the given scenario.

Regarding the last part of the referee comment, line 40 page1 - line 2 page 2 reports the study of Peruccacci et al. (2017), which indicates errors that are always less than 1 day. As already

commented in the MS (P6 L17-18; P8L17-18) and discussed also in the reply to reviewer #1, our analyses show that errors of such amount do not affect significantly threshold determination and performances. Hence, other elaborations are not needed to simulate consequences of situations similar to those reported by Peruccacci et al. (2017). The study of Peruccacci et al. (2017) reports a relatively high precision of data, because the events are selected from a larger dataset covering a whole nation (Italy), *explicitly requiring* high accuracy. This will be specified in the revised MS. Especially when dealing with regions of smaller extension (as it is more usual), the data quality requirements can be less restrictive, to retain a significantly numerous dataset. Moreover, the referee should note that we cited also Guzzetti et al. (2008), which reports (for a global dataset) a way lower precision. They reported that the vast majority of events (68.2%) had no explicit information on the date or the time of occurrence of slope failure, while for most of the remaining events only the date of failure was known; more precise information was available just for 5.1% of the events. It is out of the scope of the paper to reproduce errors occurred in specific datasets used in landslide triggering threshold assessments performed by others. Our scenarios represent a range of possibilities, respect to which real datasets may likely represent intermediate cases.

The revised MS will include some sentences aimed at making more clear what discussed above.

*Refs.*

Peruccacci, S., Brunetti, M.T., Gariano, S.L., Melillo, M., Rossi, M., Guzzetti, F., 2017. Rainfall thresholds for possible landslide occurrence in Italy. Geomorphology 290, 39–57. https://doi.org/10.1016/j.geomorph.2017.03.031

Berti, M., Martina, M. L. V, Franceschini, S., Pignone, S., Simoni, A. and Pizziolo, M.: Probabilistic rainfall thresholds forlandslide occurrence using a Bayesian approach, J. Geophys. Res. Earth Surf., 117(4), 1‑20, doi:10.1029/2012JF002367, 2012.

Vessia, G., Parise, M., Brunetti, M. T., Peruccacci, S., Rossi, M., Vennari, C. and Guzzetti, F.: Automated reconstruction of rainfall events responsible for shallow landslides, Nat. Hazards Earth Syst. Sci., 14(9), 2399‑2408, doi:10.5194/nhess-14-2399-2014, 2014.

Melillo, M., Brunetti, M. T., Peruccacci, S., Gariano, S. L. and Guzzetti, F.: An algorithm for the objective reconstruction ofrainfall events responsible for landslides, Landslides, 12(2), 311‑320, doi:10.1007/s10346-014-0471-3, 2015.

Guzzetti, F., Peruccacci, S., Rossi, M., Stark, C.P., 2008. The rainfall intensity-duration control of shallow landslides and debris flows: An update. Landslides 5, 3–17. https://doi.org/10.1007/s10346-007-0112-1

6 – The case of the Italian rainfall dataset is presented in which precipitation for the day D is collected for the 24h preceding 9am of day D. Wouldn't one use this dataset by shifting it by one day? So that precipitation of day D is between 9am of day D and 9am of day D+1? Surely there will still be some error as it still wouldn't match with the day definition, but this would probably be more meaningful.

We agree with the referee on this point. By the case of the "Italian rainfall datasets" we show what are the consequences of being unaware of the aggregation shift. Of course, if the analyzer is aware of this artifact, he would try to exploit the dataset at best, i.e. by shifting the original data as mentioned by the referee. And indeed in the conclusion this is what we want to stress in (p8 lines 29-33: "when threshold are determined from daily data, the data analyst has to be aware of possible shifts/delays in the rainfall accumulation interval, that is, if precipitation reported for a given day is the total amount occurred in a shifted period"). When corrected as the referee suggests, one would obtain low impacts. Nevertheless, we believe that the issue of shifted rainfall amounts deserves to be explicitly discussed, as is done in our MS. This because, apart from few papers (only Caracciolo et al., 2017, to our knowledge), most of the papers focused on the determination of landslide triggering thresholds in Italy (for which this shift can be present), do not report any relative correction. From this we may infer that in a significant number of studies the analyzer was not aware of the shift, since it would have been otherwise mentioned. There is no need for doing additional elaborations, as the results would be quite similar to those obtained in Fig. 7c (cf. also answer to referee #1).
More detailed discussion on these issues will be added to the revised MS.

**Nat. Hazard Earth Syst. Sci. Discuss., https://doi.org/10.5194/nhess-2017-328**

Influence of uncertain identification of triggering rainfall on the assessment of landslide early warning thresholds, by David J. Peres, A. Cancelliere, R. Greco and T.A. Bogaard.

**List of modifications related to comments by Referee #1**

Please notice that page and line numbers are those of the revised version of the MS

| Referee comment | Modifications |
|---|---|
| **General Comments**
The manuscript of Peres and co-authors entitled "Influence of uncertain identification of triggering rainfall on the assessment of landslide early warning thresholds" is an interesting well-structured and well-written manuscript that addresses a very important scientific question that is within the scope of NHESS. However, it needs some minor revisions prior to be published. | - |
| **Specific Comments**
1 - The exercise presented along the manuscript is based on synthetic data, which are easier to control and monitor. However, the exercise has the drawback of reporting a single ideal slope. So, there is also a matter of scale when we compare the obtained results with most rainfall thresholds reported in literature that were built to be applied and interpreted at the regional scale. May be this is not enough discussed along the manuscript. | P3 L11-14 The application to a hillslope of definite characteristics enables us to isolate the impact of the uncertainty in triggering rainfall identification; regional determination of thresholds do contain also factors of uncertainty related to the heterogeneity of landslide characteristics; the assessment of this combined uncertainty is out of the scope of our present analysis. |
| 2 - Within the simulation of uncertainty in triggering instant and the reporting of the landslide, authors establish the 'Observer's day' as lasting from the 6pm of Day D-1 to 6pm of the Day D. The explanation of this option is not clear. Although the reporting of a landslide in newspapers is usually delayed in relation to the actual triggering instant, the information about the timing of triggering may be quite precise namely in those cases where landslide generated severe human and/or economic damages. Apparently, this was not considered in the definition of the 'Observer's day'. | P4 L 1-2 […] this choice is an attempt to resemble usual working hours, and the fact landslides occurring by night may be reported the morning after

P4 L14-15 The two parameters, $T_O$ and $T_A$, can be set to simulate a range of scenarios, for which real situations may represent intermediate cases

P8 L 34- 38To this aim, we have investigated the effect of a set of hypothesized scenarios of landslide information retrieval and interpretation which can induce errors in the identification of instants of landslide occurrence. Moreover, we have analysed how the impact of reasonable scenarios may vary in dependence of rainfall aggregation |

| | (hourly or daily), and of rainfall event identification criteria. Real situations may be a mixture of the considered scenarios, and thus the impacts are presumably intermediate between the ones hypothesized. |
|---|---|
| 3 - Quite interesting, figures 6 a), 6 b) and 7 c) are very similar. Comparing figure 6a) and 6b) one can conclude that working at the daily scale the knowledge of exact timing of the landslide triggering is not essential, providing the reporting Day (D) is correct. In addition, when the daily rainfall depth is measured form 09:00 AM to 09:00AM it is clear that most of the rain that falls in the day D will be registered in the day D+1. Therefore, it is normal that threshold (c) corresponding to Scenario RS2 (Day D+1) in figure 7 is virtual similar to the Scenario RS1 (Day D) and RS0 (actual triggering instant) in figure 6. In the opinion of the reviewer, this topic should de discussed more in detail in the paper. | P1 L 19-21 The analysis shows that the impacts of the above uncertainty sources can be significant, especially when errors exceed one day or the actual instants are after the erroneous ones.

P 7 L 24-27 There is, however, the possibility that the errors due to rainfall aggregation and reporting landslide time interval compensate for each other, as in the case of scenario RS2 (delayed reporting of landslides), Fig. 7c (notice that this plot is similar to Fig. 6b). If the analyser is aware of the rainfall-aggregation shift, then he should correct as much as possible for this error – in this specific case, by shifting the entire daily rainfall dataset one day forward.

P9 L13-14 From our analysis no significant impacts seem to be induced by the use of daily data; however, it is of fundamental importance to check, and correct where possible, for the presence of delays in the rainfall accumulation interval |
| 4 - Although this information is contained on Figures 8 and 9, the equations of thresholds could be provided in a summary table, allowing for a more easy comparison. | Figures 8 and 9 of previous MS have been removed and replaced by Tables 6 and 7 showing the same information |
| 5 - When performing the exercise for the daily scale that is summarized in figure 6 and 7, a contradiction exists, between figures and text (page 6 line 35) on the assumed Smin. In figure caption it is referred Smin = 0 mm whereas in text is referred Smin = 5 mm. | P7 L16 Figure 6 shows the results of calibration obtained with correctly-aggregated daily rainfall data and $s_{min} = 0$ and $u_{min} = 1$ day |
| 6 - In figure 10 authors present the "correct thresholds". However, it is not given the information on the considered Umin and Smin parameters. | The missing information was added to Figure's caption (Fig.8 in the revised MS) |
| **Technical corrections**
In figure 2, the time scale should be respected. The position of 6pm in Day D and Day D-1 is not correctly scaled. Add the notation RS0 in figure 2. | Figure 2 has been corrected as suggested |

| | |
|---|---|
| Figure 3 The aggregation of data within figure 3 should be clearer. Rain gauge D+1appear two times; why? The total amount of rain measured on calendar days and raingauge days is not the same. Authors should acknowledge this difference and explain why. | The figure has been improved and corrected. Caption has been integrated with more information: Figure 1: Aggregation of rainfall data from hourly to daily time scale: daily rainfall depths on the top row result from correct aggregation; those on the bottom row from shifted aggregation, as occurs for the Italian Hydrological Bulletins (Annali Idrologici). The shift is due to manual collection of data in early decades of operation of the monitoring network; the presence of the shift is still continued, in spite of installation of automatic rain gauges, to preserve homogeneity of the entire historical time series. |
| Table 3 Some rainfall event identification instead of Some event identification. | Fixed |
| Reference of the paper of Nikolopoulos et al needs to be corrected in reference list. | Fixed |
| Page 2. Line 26 Rodriguez-Iturbe et al., 1987a, 1987b instead of Rodriguez-Iturbe et al., 1987; Rodríguez-Iturbe et al., 1987. Introduced a and b in the reference list. | Fixed |
| Page 2, line 31 Baum and Godt, 2010, instead of Baum et al., 2010 ? | Fixed |
| Page 3. Line 7 Schilirò et al., 2015a, 2015b, 2016; instead of Schilirò et al., 2015,2016; Schilirò et al., 2015; | Fixed |
| Page 3, line 38 Guzzetti et al 1997, 1998 are missing in reference list. | Correct citation is Guzzetti et al 2007, 2008 |

**Nat. Hazard Earth Syst. Sci. Discuss., https://doi.org/10.5194/nhess-2017-328**
Influence of uncertain identification of triggering rainfall on the assessment of landslide early warning thresholds, by David J. Peres, A. Cancelliere, R. Greco and T.A. Bogaard.

**List of modifications related to comments by Referee #2**

Please notice that page and line numbers are those of the revised version of the MS

| Referee comment | Modifications |
|---|---|
| **GENERAL COMMENTS**

In this manuscript the authors investigate the effects of uncertain knowledge of the timing of landslide occurrence on the definition of intensity duration rainfall thresholds. The study is based on synthetic rainfall data and virtual landslide events. Thresholds are defined using the True Skill Statistic as optimization criterion. The work is carried out for one ideal slope in the Peloritani Mountains in Sicily (IT). Overall the paper is well written, with a clear structure and objective. I believe it could benefit from some more elaborations on some of the aspects presented, mentioned here below. I recommend minor revisions before publication on the journal. | |
| **SPECIFIC COMMENTS**
1 – On the line of what already mentioned by Anonymous Referee #1, the study is purely focused on one ideal slope and synthetic data. The authors could discuss how this might make the results transferable to a real situation, when regions are considered and heterogeneities come in to play. This with respect especially to the difference in the scale and the use of virtual landslides. | P3 L11-14 The application to a hillslope of definite characteristics enables us to isolate the impact of the uncertainty in triggering rainfall identification; regional determination of thresholds do contain also factors of uncertainty related to the heterogeneity of landslide characteristics; the assessment of this combined uncertainty is out of the scope of our present analysis. |
| 2 – The authors should report the total number of landslides as well as of non-triggering events considered. While this probably changes with the different parameters for the definition of the events, it would be useful to give an idea of the "robustness" of the results, that is whether the change of just few events among different scenarios would affect or not the threshold. Although the TSS considers both triggering and non-triggering events, the less the triggering events the more their relative importance on the definition of the threshold. | P3 L27-28 Table 2 shows some characteristics of the 1000-year long synthetic databases, which do not change among the different scenarios illustrated in the following section.

P6 L19-21 One advantage of the TSS is that it includes all the entries of the confusion matrix, and thus its maximization yields thresholds that result in a good trade-off between correct and wrong warnings/non-warnings. |

| | |
|---|---|
| 3 – The authors could elaborate more on how the threshold was defined, as the results are difficult to explain without this information. An example is the change going from the case shown in Figure 5a to 5b. The "two rainfall events shifted to a duration of 1 h" (line 18-19 page 6 in the text) cannot be responsible for the lowering of the threshold intercept or slope as they are not correctly captured by the threshold but are "missed". So either some other triggering events changed causing the decrease of the threshold or the threshold shouldn't have changed. All this is true unless the authors gave somehow weight also to the distance from the threshold. If being just below the threshold or well below the threshold makes a difference in the TSS, then yes those points could be responsible for the change and you should ignore this comment, but it would be helpful if the method would be explained.

4 – It seems that in general the points in the ID plane always move down (or left) in all the different scenarios. One would expect that sometime the landslides occur during intense rainfall storms and therefore including some extra hours actually could increase the intensity and duration. | P6 L34 –L40 The presence of small delay reporting errors (RS1), has little impacts on the position of triggering rainfall points (Fig. 5b), which in general are shifted slightly down along the intensity axis; this is related to the higher durations produced by positive errors in triggering instants, combined with an induced decrease of mean rainfall event intensities – a general behavior exhibited by extreme events (cf. the negative slope of well-known rainfall intensity-duration-frequency curves, see Bogaard and Greco, 2018). Only two rainfall events (the 2.5 % of triggering events) are highly-impacted, being moved to a duration of 1 hour. The latter and mainly the former effect, contribute to slightly flatten the threshold for TSS maximization (decrease of $\beta$ το 0.7) |
| 5 – The authors could explain better how the different scenarios are then used and corresponding triggering events selected. In fact, the scenarios are explained very well, but it is unclear how the events are then constructed. Is $e_i$ randomly selected for each virtual landslide within the range defined for each scenario? Are then the results shown only one possible realization? Or is the wrong timing always fixed to Ta (that is always midnight, either 0, 24 or 48)? In other words, is the triggering event always the one happening at midnight or the last one that happened just before then? That wouldn't be a very realistic case because one would either try to find out at least whether it was morning or afternoon, or choose the most intense event within the day (which would then result in an overestimation of the threshold, but probably would still better than taking midnight rain) or choose the typical timing of landslides. Also for an available database, not for all entries | P4 L 8-9 These errors are implicitly random, since though $t_i'$ are deterministically chosen, the actual instant $t_i$ varies in an aleatory fashion according to rainfall time history.

P5 L12-14 Automatic procedures have the advantage of being objective and reproducible, and thus more scientifically sound than subjective judgment (Melillo et al., 2015; Vessia et al., 2014); nevertheless, algorithms are suitable to reproduce the latter with a certain level of fidelity (Berti et al., 2012).

P4 L19, L23, L28 "random in the range" has been added

P2 L2-3. In their analysis, only information with an accuracy at least of one day was retained from the larger available dataset. Still for this trimmed dataset, triggering instants were available with high precision (minute or hour) only for the 37.3% of the data, being the day or part of it available for the majority (27.6% and 35.1%, respectively). |

| | |
|---|---|
| timing or at least part of the day would be unknown (for the example you report in line40 page1 to line2 page2, only 27.7% of the cases would fall in this case, of only day know) | P2 L19-25 We then fictitiously introduce errors in the triggering instants and in the rainfall series based on hypothetical scenarios of landslide data retrieval and analysis, and analyse the implications on the accuracy of ID thresholds. Quality of information available in real datasets is generally intermediate of that corresponding to the hypothesized scenarios. These scenarios are combined with different criteria for event rainfall identification, and different aggregations of rainfall data (hourly and daily, and daily in the presence of a shift due to manual collection of data), so the effects of these other two sources of uncertainty are analysed as well (items *i)* and *ii)* of the above list).

P4 L 14-15 The two parameters, $T_O$ and $T_A$, can be set to simulate a range of scenarios, respect to which real situations may represent intermediate cases

P8 L37-38 Real situations may be a mixture of the considered scenarios, and thus the impacts are presumably intermediate between the ones hypothesized. |
| 6 – The case of the Italian rainfall dataset is presented in which precipitation for the day D is collected for the 24h preceding 9am of day D. Wouldn't one use this dataset by shifting it by one day? So that precipitation of day D is between 9am of day D and 9am of day D+1? Surely there will still be some error as it still wouldn't match with the day definition, but this would probably be more meaningful. | P7 L 26-27 
[revised manuscript text omitted]
|  | 24 | 5 | 0.6 | 28 | 0.7 | 0.73 | 40 | 0.8 | 0.89 | 30 | 0.7 | 0.29 | 21 | 0.7 |